# Impact of *Saccharomyces cerevisiae* on the Field of Single-Molecule Biophysics

**DOI:** 10.3390/ijms232415895

**Published:** 2022-12-14

**Authors:** David A. Ball, Binta Jalloh, Tatiana S. Karpova

**Affiliations:** CCR/LRBGE Optical Microscopy Core, National Cancer Institute, National Institutes of Health, Bethesda, MD 20852, USA

**Keywords:** SMT, protein dynamics, transcription, HaloTag, SNAP-Tag, chromatin remodelers, transcription factors, replisome, yeast, *Saccharomyces cerevisiae*

## Abstract

Cellular functions depend on the dynamic assembly of protein regulator complexes at specific cellular locations. Single Molecule Tracking (SMT) is a method of choice for the biochemical characterization of protein dynamics in vitro and in vivo. SMT follows individual molecules in live cells and provides direct information about their behavior. SMT was successfully applied to mammalian models. However, mammalian cells provide a complex environment where protein mobility depends on numerous factors that are difficult to control experimentally. Therefore, yeast cells, which are unicellular and well-studied with a small and completely sequenced genome, provide an attractive alternative for SMT. The simplicity of organization, ease of genetic manipulation, and tolerance to gene fusions all make yeast a great model for quantifying the kinetics of major enzymes, membrane proteins, and nuclear and cellular bodies. However, very few researchers apply SMT techniques to yeast. Our goal is to promote SMT in yeast to a wider research community. Our review serves a dual purpose. We explain how SMT is conducted in yeast cells, and we discuss the latest insights from yeast SMT while putting them in perspective with SMT of higher eukaryotes.

## 1. Introduction: Why Single Molecule Tracking?

Molecular biology provides static snapshots of cellular processes, while quantitative fluorescence microscopy allows observation of their dynamics in live cells. Based on ChIP (Chromatin Immunoprecipitation), protein complexes appear to be built in an orderly, sequential, linear fashion, and the assembled complexes appear to persevere on the time scale of minutes until the task is completed (see example in [1] for mammalian cells and in [2] for budding yeast). FRAP (Fluorescent Recovery After Photobleaching) and FCS (Fluorescence Correlation Spectroscopy) microscopy assays led to a surprising paradigm shift demonstrating that the binding of proteins to their molecular targets and assembly and function of a great majority of cellular complexes are dynamic on the scale of seconds. Rapid dynamics for Transcription Factors (TFs) were first revealed in mammalian cells [3] and later in yeast [4,5,6]. In mammalian cells, FRAP data were corroborated by FCS; FCS and FRAP parameters were reconciled [7]. FCS was never implemented in yeast due to its small size.

However, FRAP and FCS results are averages for the whole population of molecules in the cell, and information on the molecular behavior must be extracted from FRAP and FCS by complicated mathematical modeling that lowers the sensitivity of the methods. Direct information about the activities of individual molecules may be obtained by SMT. By following single molecules labeled with photostable fluorescent markers, SMT provides parameter estimates of protein binding rates, and reveals protein mobility states and mechanisms of molecule search for specific targets.

By SMT, the in vivo dynamics of TFs binding to DNA were first observed in mammalian [7,8,9,10] and yeast [11] nuclei. Later, SMT was applied to specific sites in yeast and revealed rapid inter-dependent cycling of TF Ace1p and chromatin remodeler RSC at the promoter of the *CUP1* gene [12] and dynamic binding of TF Gal4 to the promoter of *GAL10* [13]. Other groups investigated the dynamics of DNA binding for replisome components [14], for histone H2A.Z and remodeler *SWR1* [15], for the RNA Pol II and components of the Preinitiation Complex (PIC) [16], and for chromatin remodelers [17]. Nowadays, fast kinetics for the great majority of nuclear proteins are generally accepted. 

SMT observations led to the following questions: what is the physiological relevance of these rapid dynamics? How do the fast dynamics of individual components affect protein complexes? How do these dynamics translate into the downstream cellular processes? How do the proteins search for their target sites, such as DNA, RNA, or nuclear bodies? How are protein dynamics regulated? Some of these questions have already been answered, but there is still a long way to go before a complete understanding can be reached.

SMT in live cells was pioneered in the field of transcription; thus, most of the papers published and discussed in this review, with few exceptions, are about the biophysics of gene expression, and we hope that our review will contribute to the expansion of SMT techniques to other cellular processes, as the technology should work the same regardless of the protein function.

We summarize current research in SMT for yeast. However, we expanded the discussion to relevant cases in mammalian cells, *Drosophila*, and bacteria. The data for a single experimental system will be useful only for this system if they are not discussed in the context of the overall research. Comparison of yeast SMT with other eukaryotes and prokaryotes allowed us to extract general features for the behavior of transcription factors and to define yeast-specific features. More studies are required to determine general features and those specific to yeast for other cellular factors.

## 2. Know-How of Yeast SMT

To analyze the mobility and binding of single molecules, we need to (a) label molecules in live cells with a photostable dye, (b) ensure that each molecule of the protein is labeled by just a single organic dye molecule, (c) record molecules on an imaging setup that enhances the signal to noise ratio (SNR), (d) identify molecules and track them in the recorded time-lapse movies, and (e) convert tracks of the molecules into biophysical parameters.

### 2.1. How to Label Yeast for SMT?

For yeast, due to its small size, we recommend imaging diploids because diploids have bigger cells and nuclei. Proteins of interest may be labeled by fusing them to HaloTag or SNAP-Tag [18,19]. Specially designed cell-permeable HaloTag and SNAP-Tag ligands associated with organic photostable dyes covalently bind tagged proteins of interest in vivo. Wild-type yeast strains are hard to label due to low retention of the xenobiotic compounds, which are constantly pumped out of the cells by ABC-MDR transporters. The best way to achieve efficient staining is to knock out just one of the three transporters, *PDR5* [11]. A visible improvement in staining is observed in *pdr5* homozygous Knock Out (KO) diploid cells in comparison to wild type (Figure 1A).

Ball et al. demonstrated that knocking out *pdr5* does not affect the binding of Ace1p to promoters of the *CUP1* gene. Conversely, electroporation is not recommended for HaloTag and SNAP-tag labeling. It is a harsh treatment that may upset sensitive processes one might want to study; for instance, electroporation disrupts the normal cycle of Ace1p binding to activated promoters of *CUP1* [11]. This underscores the importance of physiological controls to determine if the labeling strategy used is benign and has no adverse effects on the cells.

Photostability of the organic fluorescent dyes used for HaloTag and SNAP-Tag ligands is very important for SMT. The biggest challenge in SMT is the photobleaching of the observed fluorophore molecule. For the bound molecules, photobleaching curtails the quantification of binding. The best current dyes (JF series) were developed at HHMI (Janelia) [20]. New and more photostable dyes are constantly being developed. Tracks of the individual molecules may be visualized as bright lines on a kymograph, which represent projections of the field of view along the X or Y axis over time. Molecules may appear, stay, and eventually disappear during observation due to diffusion or photobleaching (Figure 1B, Left). The survival distribution is derived from tracking individual molecules; it characterizes the time that observed molecules are in focus for the population (See schematics, where ten molecules observed in the first frame gradually disappear: Figure 1B, Right). Histone proteins stably bound to nucleosomes are one of the most immobile nuclear proteins; therefore, their dwell time is curtailed not by mobility, but mainly by the photobleaching of their fluorescent marker. Survival plots of histone H3-HaloTag fusion labeled with JF_646_ or the newer JFX_650_ ligands demonstrate that the latter survive photobleaching better: the blue curve of *HHT1*-HaloTag/JFX_650_ is consistently higher than the red curve for the JF_646_ label, indicating longer survival for JFX_650_, i.e., more molecules are present in the field of view for JFX_650_ at any time point. At the end of the observation, all molecules of JF_646_ are photobleached, but a sizeable fraction of JFX_650_ is still present (Figure 1C).

### 2.2. How to Image Yeast for SMT?

#### 2.2.1. Imaging Systems

Single molecules emit weak signals, which present a challenge for the detection and tracking of live cells above the auto-fluorescence background. Long-term imaging of single molecules with wide-field illumination may present a problem; for single molecules to become visible, it requires pre-bleaching of the field-of-view to reduce the background noise [15,16]. Stably bound molecules are photobleached faster than mobile molecules, and thus pre-bleaching may lead to biases toward mobile molecules. The four techniques best suitable for SMT are HILO (Highly Inclined and Laminated Optical sheet), LLS (Lattice Light Sheet), Orbital Tracking, and MINFLUX (MINimal photon emission FLUXes) (Figure 2A).

HILO and LLS substantially reduce out-of-focus fluorescence by using a thin sheet of light for excitation. To increase SNR and thus enable SMT, a laser sheet with a thickness of ~5 μm is produced by tilting and flattening a laser beam on a custom microscope with HILO illumination [21]. The tilted beam at the sample plane causes the Gaussian (bell-curve-shaped) profile of the laser to be aligned in the axial direction rather than radially, thus reducing the excitation of fluorophores that are out of focus. The depth of field for the 100X/1.45 NA objective is ~400 nm; thus, although the entire yeast cell is illuminated, fluorescence from only one-third of the yeast nucleus is collected, and not all fluorescent particles may be visible at the same time. The inclined light sheet produces an illumination gradient in the field of observation, as discussed in [22]. For large mammalian nuclei, this gradient results in different levels of photobleaching in different parts of a single nucleus, and thus affects the correction procedures for photobleaching. However, our group established experimentally that even with the illumination gradient, each individual yeast nucleus is exposed to a uniform amount of light due to the small size of the yeast cells. HILO illumination limits the depth of penetration into the specimen above the coverslip surface to achieve reasonable SNR. The typical precision of detection of molecular positions in such experiments is ~40 nm. In yeast, HILO was applied to SMT by many groups, including [12,13,23]. Most commercial wide-field microscopes capable of performing TIRF (Total Internal Reflection Fluorescence) microscopy can be used for HILO imaging. 

A commercially available LLS microscope relies on using separate objective lenses for illumination and image collection. The illumination in LLS is produced by a Bessel-shaped, rather than Gaussian, laser beam formed into a lattice by the use of a spatial light modulator. This pattern is dithered perpendicular to the detection objective, which results in a thin sheet of illumination beyond which the areas are not irradiated [24]. The thickness of illumination is matched to the depth of focus of the detection objective—both are approximately ~1 μm. The sample is mounted on a fast piezo-electric stage, which allows sample scanning to perform 3D imaging. LLS has no limitations on the distance from the coverslip, which is important for tissue imaging. However, for yeast, which is always localized at the coverslip, this feature of LLS brings no advantage. The resolution of LLS in time and space is similar to a wide-field illumination, but the lower radiation dose supplied by LLS reduces cell photodamage and offers the possibility to perform tracking single molecules in 3D with reasonable time resolution [10]. However, LLS has yet to be applied to yeast. 

Common 2D SMT imaging technology is limited by photobleaching and sites going out of focus. Thus, it is not well suited for simultaneous observation of two processes occurring on different time scales requiring long-time imaging in 3D and fast time sampling in two channels. Orbital Tracking allows two-channel observation of fluorescent protein binding in 3D to moving fluorescent target sites [25]. In the Orbital Tracking technique, laser beams for both channels are combined for simultaneous observation; beams make circles (radius ≈ 100 nm) positioned ~150 nm above and below the fluorescently-labeled area of interest. One of the two color channels is used for monitoring the intensity and position of the target site over time; this information is used for adjusting the center of the circles in 3D. The second color channel is used for the detection of a tagged protein, such as a TF, when it is bound to the target site, with a minimal time resolution of ~250 ms. The resulting two-color signal time-courses are typically analyzed using cross-correlation functions. Although it is not capable of following diffusing particles detached from target sites, Orbital Tracking registers binding events; thus, it has been used to measure the residence times of bound single molecules at a specific location, such as at a Transcription Site (TS). In yeast, orbital tracking was applied to simultaneous imaging of nascent *GAL10* mRNA and TF Gal4 [13]. Several approaches were described for achieving Orbital Tracking [26], and commercial options for microscopy systems are available.

MINFLUX is a new method of SMT that promises unprecedented localization precision down to 2 nm and a temporal resolution of 1 ms [27]. On a MINFLUX microscope, a donut-shaped laser beam is positioned in a hexagonal pattern around each detected molecule. The position of the molecule is triangulated based on the intensity recorded at each of the six corners of the hexagon. Tracking of the molecule is achieved by constantly moving the pattern to be centered on the position of the molecule found in the previous time step. This brand-new technique has not yet been applied to yeast; however, MINFLUX tracking of kinesin in live mammalian cells has recently been reported [28].

In addition, 2D or 3D SMT with super-resolution in live mammalian cells was performed on custom-built systems, such as Target-locked 3D STED [29], a multi-focus system [30], a system with multiplexed PSF engineering [31], and astigmatism imaging system [32]. However, commercially available instruments are the best option for most researchers doing SMT. 

#### 2.2.2. Detection of Molecules

Efficient tracking requires a small number of labeled molecules per cell. A low concentration of fluorescent particles (1–2 per nucleus) may be achieved in two ways. First, HaloTag and SNAP-Tag fusions may be labeled sparsely (using very low concentration), as shown in Figure 1A, top, and [12,13,15]. Second, a low number of fluorescent particles may be achieved by controlled photoactivation of photoinducible fluorophores [14]. Photoactivation allows multiple rounds of imaging due to repeated cycles of photoactivation after the first set of activated molecules are photobleached (Figure 2B, bottom).

Live yeast cells may be mounted onto LabTek coverglass chambers under agarose pads with CSM; yeast cells are immobilized and slightly flattened by the agarose pad. Nonfluorescent media instead of CSM may reduce autofluorescence [33]. However, other authors reported mounting yeast cells in liquid CSM on coverslips carefully cleaned and flamed to reduce autofluorescence and then coated with Concavalin A to immobilize the cells [14,15].

#### 2.2.3. Tracking of Molecules

Images are collected sequentially with defined time intervals. The brightness of the molecules defines the minimal exposure allowing visualization of the molecule above the image noise. Fast (~10–12 ms) acquisition with continuous illumination with a high-power laser is applied if the focus is mainly on diffusing and short binding events. Slow acquisition with long exposures (≥200 ms) interspersed with dark periods is applied if the focus is on long-binding events. 

In most of the studies performed in mammalian or yeast cells, molecules are tracked in the whole cell or nucleus. For most proteins, multiple binding sites with a range of affinities exist in the cell. Therefore, the binding parameters are an average of multiple activities on multiple sites for the same protein. More information may be obtained if SMT is performed at a specific location in the cell, where a protein of interest is known to perform a specific function. Specific locations may be labeled by fluorescent markers in a different channel. This label should be readily distinguished from individual marker molecules, moving by diffusion, or bound ectopically; thus, clusters of fluorescent molecules are usually utilized for the unambiguous detection of a specific site.

In yeast strains heterozygous for Ace1p-HaloTag/JF_646_ and Ace1p-3xGFP, the former was used for SMT, and the latter for visualization of the target sites at a specific *CUP1* promoter [12]. The *CUP1* promoter contains four binding motifs for the TF Ace1p. Four molecules of Ace1p-3xGFP bound to a single promoter can be detected at high exposure but cannot be imaged repeatedly. However, the *CUP1* locus consisting of ~10 tandem copies of the same gene provides a cluster of ~40 binding motifs, which may be observed for a prolonged time by binding of the 3xGFP-tagged Ace1p. Tracking of RNA Polymerase II (Pol II) can be performed at *CUP1* promoters visualized by the Ace1p-3XGFP (Figure 2C) or at *CUP1* TS, which can be identified by the stem-loop approach by inserting a reporter into the tandem array of *CUP1* genes (Figure 2D). Another TF, Gal4, was tracked at the *GAL10* locus and labeled for nascent mRNA by the stem-loop approach [13]. It is hard to detect binding to specific sites because only a very small fraction of the appropriate TF molecules are fluorescently labeled for SMT, and thus most of the binding events involve unlabeled molecules. 

### 2.3. How to Analyze Single Molecule Tracks?

As a first step, molecular tracks must be generated from images acquired with the preferred microscopy instrument. Images are typically filtered with a Gaussian bandpass filter to highlight molecules above the background (Figure 3A).

Assembled track data may be analyzed by Survival Distribution and Mean Squared Displacement (MSD) (see the latest examples in [34,35]).

#### 2.3.1. Analysis by Survival Distribution: Bi-Exponential Fit

The survival distribution is normalized to the bound fraction calculated as (number of bound particles) × (number of time frames the particles are bound)/(total number of detected particles in all movie frames) (Figure 1B). Normalized values are divided by the total number of particles at *t*_0_. The survival time distribution is traditionally fit to exponential decay. The first point of the fitted survival curve supplies the information on the ratio of bound molecules to total (bound and diffusing) molecules, *C_eq_*. The simulated survival distribution of Figure 3B (left) indicates that bound molecules constitute 46% of the total population. The population of bound molecules may contain sub-states. Fitting by a bi-exponential decay results in two distinct sub-states for bound molecules: molecules with short residence time, interpreted as binding to non-specific targets (searching), and molecules with long residence time, interpreted as binding to specific regulatory motifs. The validity of these assumptions was tested in several papers in mammalian [22,36] and yeast cells [12,16,17]. Typically, proof that the long residence binding is specific requires conditional inactivation or deletion of the DNA-binding domain. On fluorescently labeled specific promoters, binding to specific motifs may be identified by comparing the tracking of the TF over sites with specific motifs and sites with no specific motifs. As example, in Mehta et al., the authors tracked Ace1p at specific sites in the promoters of *CUP1* and at the array of lacO and demonstrated that the long-residence fraction practically disappears at the lacO sites [12]. As a benchmark for non-specific binding, freely diffusing nuclear HaloTag may be used [16]. However, for DNA-binding proteins, a better benchmark is a protein that is conditionally incapable of specific binding while still retaining the ability for non-specific binding. A great example of this is Ace1p, which has only five specific sites for binding in the genome, and this binding requires the association of Ace1p with Cu^2+^ ions. In the absence of copper, TF Ace1p may bind only nonspecifically to targets in the nucleoplasm [11]. 

Bi-exponential distributions provide estimates for the residence times, *τ_NS_* and *τ_S_*, which characterize the average time that nuclear factors remain bound to non-specific and specific targets on chromatin, respectively. The fitting also reveals the fractions of molecules in each state (i.e., *F_NS_* and *F_S_*). From the fractions in each bound state, one may estimate the average number of chromatin sites sampled by a molecule before it finds a specific motif, *N_trials_*,
$$N_{trials}=\frac{F_{S}+F_{NS}}{F_{S}}$$

The search time (the time nuclear factors spend in search of the specific sites) may be estimated (Figure 3B, Left): $$\tau_{search}=\frac{1-C_{eq}}{C_{eq}}\tau_{res},$$
where *τ_res_* is the average of the non-specific and specific binding,
$$\tau_{res}=\frac{F_{NS}}{F_{NS}+F_{S}}\tau_{NS}+\frac{F_{S}}{F_{NS}+F_{S}}\tau_{S},$$

Changes in *C_eq_* may result from: (a) a change in the availability of the TF, *C_free_*, (b) a change in the TF dissociation rate, *k_off_*, (c) a change in the availability of the specific motifs, *S_eq_*, or (d) a change in the TF association rate, *k_on_*. *S_eq_* indicates how many specific motifs in the promoter are free from the nucleosomes and accessible to TFs, and *k_on_* is dependent on the size of the area for the search [37]. Directly from SMT data, one can estimate only “pseudo ‘on’ rate” ($k_{on}^{*}$):$$\tau_{search}=k_{on}^{*}{}^{-1}={\left(k_{on}S_{eq}\right)}^{-1}.$$

To extract *k_on_*, the availability of the specific motifs, *S_eq_,* is required. *S_eq_* may be available for some TFs from previously published estimates. Alternatively, *S_eq_* may be estimated from the fraction of non-activated specific loci observed by RNA smFISH (single molecule Fluorescence In Situ Hybridization) [12].

#### 2.3.2. Analysis by Survival Distribution: Power Law Fit

The survival curve may follow not a bi-exponential, but a Power Law distribution (Figure 3B, Right). A Power Law distribution indicates that binding events occur at multiple sites with a wide range of affinities, which are impossible to divide into distinct populations with specific and non-specific binding. Fractions of diffusing and bound molecules may be estimated from the observed molecular distribution. No specific residence times can be estimated from Power Law fits. For the majority of mammalian nuclear factors studied by SMT, multiple binding sites with vast variations in affinity exist. Thus, the kinetics of the search and binding to different sites in different chromatin environments must also be complex and better described by the Power Law.

The length of time that the observed molecules appear to remain stationary depends on binding kinetics as well as on photobleaching, and therefore the raw survival distribution (Figure 3C, black points) must have the photobleaching contribution removed prior to analysis. Recently, a rigorous study showed that the outcome of quantification strongly depends on the specific photobleaching correction that is used [38]. Existing approaches to correct photobleaching are based either on intrinsic correction from the collected molecules themself [7,12] or a correction based on a relatively immobile protein, such as histones [16,39]. To illustrate the effects of the various photobleaching correction methods, we performed photobleaching correction for the same histone H3 data intrinsically, as shown by the red triangles in Figure 3C; by the whole (same) histone population, as shown by the blue circles; or by the immobile fraction of histones, as shown by green asterisks. 

Intrinsic correction was justified by observations that molecules with different mobility are photobleached at different speeds. Thus, estimates of the bleaching rate from the number of particles found over time for a tracked protein were thought to be more reliable than estimates based on the survival curve of a different, stable protein. It was observed, though, that intrinsic correction does not result in sufficiently long residence times for stably bound proteins such as histones. Therefore, the intrinsic correction underestimates the photobleaching of the sample. More problematic was the observation that the estimates of binding parameters for the same protein are vastly different depending on the time interval used in the time-lapse imaging. Photobleaching correction based on histones does provide long residence times for stably bound proteins. In fact, correction of the histone survival distribution in this manner results in a perfectly flat curve, indicating that all molecules are bound for at least the length of the acquisition time. However, inspection of the time-lapse movies of histones reveals that there are fractions of unstably bound molecules. Thus, correction by the total histone survival artificially removes mobile subfractions by overestimating the photobleaching of the sample. As a compromise between the above two extremes, [38] suggested correction by the most immobile fraction of histones obtained from tri-exponential fits of the histone survival distribution. This correction allows the reconciliation of the estimates from different time-lapse experiments. Applied to histones, it results in a distribution that plateaus (indicating a stably bound population), while retaining the unstable binding component (Figure 3C, green asterisks). This corrected survival curve best matches the H3 visible behavior.

In a real experiment, one may encounter more complex fitting. For RNA Pol II in yeast, after correcting for photobleaching with the immobile fraction of histones, we observed that the data are best fit with a combination of Power Law and exponential decay (Figure 3D). This indicates that three states of motion are present, characterized by diffusive, short-bound, and long-bound events. Fitting to a function that is a mixture of Power Law and exponential distributions can still give information on the fraction of bound and diffusing molecules from the first point in the fit (as described above for bi-exponential fitting), the fractions of short- and long-bound molecules (corresponding to exponential and Power Law components, respectively), and the average residence time for the short-bound state. In this case, however, we cannot estimate the average residence time for the long-bound population as its distribution follows the Power Law. 

#### 2.3.3. MSD Analysis

The MSD curve may be estimated in individual tracks by measuring the distance traveled by a molecule at different time intervals (Figure 3E). The data for individual tracks are averaged to obtain the MSD for the whole population (Figure 3F). 

Typically, to extract MSD estimates, fast tracking with short time intervals is applied. Fitting the initial, linear portion of the MSD curve for an individual track gives an estimate for the apparent diffusion coefficient (D). The distribution of the logarithm of extracted diffusion coefficients for the individual tracks can then be fit by Gaussian, bi-Gaussian, or tri-Gaussian models to extract the sub-fractions of bound, intermediate, and unbound particles [15]. Figure 3G shows a simulated distribution of the logarithmically-transformed D exhibiting three sub-fractions. For those sub-fractions, their ratio in total molecular population and their specific D may be estimated.

The MSD may also be approximated by a Power Law (i.e., MSD(t) = t^α^) to determine the type of motion encountered. Brownian motion (normal or pure diffusion) is characterized by α = 1, while α > 1 indicates facilitated motion (super-diffusion), and α < 1 indicates confined motion or sub-diffusive processes [40]. For particles that exhibit sub-diffusion, one can calculate the area to which the mobility of the given protein is limited, which is characterized by the radius of confinement *R_c_* [41],
$$MSD\left(t\right)=R_{c}^{2}\left(1-e^{\frac{-4Dt}{R_{c}^{2}}}\right).$$

The maximal size of such a confinement is limited by the size of the compartment, for example, the size of the nucleus for the nuclear proteins (*R_c_* = 1 μm for diploid nucleus). For all tracks obtained from yeast RNA Pol II, we observed α = 0.35, indicating sub-diffusion, and *R_c_* = 0.3 μm, which reveals that Pol II is confined to a smaller area of the nucleus (Figure 3F). Interestingly, although we applied a very different time interval for imaging, used a different dye, and imaged the cells on a different type of microscope, the *R_c_* = 0.31 μm of Pol II in our experiments matches previously reported data if it is recalculated for the whole population of Pol II molecules. Previously, *R_c_* = 0.13 μm for the bound sub-population, *R_c_* = 0.5 μm for the unbound sub-population, and the fraction of the bound sub-population as 50% were reported [16]. Therefore, in these experiments, total *R_c_*___ = 0.5 ∗ 0.13 + 0.5 ∗ 0.5 = 0.315 µm. Thus, it appears that Pol II uses a sub-diffusive process to explore the yeast nucleus, with a typical sampling volume of 0.113 μm^3^.

#### 2.3.4. Software for SMT

SMT analysis software should provide tracking of the particles and parameter estimates for binding. Different software may be used for tracking and for further analysis. Our group uses MatTrack, which performs both tracking and analysis. The software is a MATLAB package that allows the use of various tracking algorithms for linking particles from frame to frame. The software was initially developed to extract estimates of residence times via fitting of survival distribution with the intrinsic photobleaching correction [42]. It has since been expanded to include stable histone photobleaching correction [38] for residence time analysis and to analyze motion via MSD.

Other groups prefer to use separate software to perform the tracking and analysis. There are several open-source options for tracking single molecules, which include uTrack [43] and the FIJI plug-in TrackMate [44], which both utilize a Linear Assignment Problem (LAP) algorithm for linking particles from frame to frame. DiaTrack performs linking using a graph analysis [45], and SLIMfast uses a multiple-target tracing (MTT) algorithm [46]. For a comprehensive comparison between various tracking methods, the reader is directed to the review by Chenouard and colleagues [47].

Regardless of the methods used to obtain tracks of single molecules, the next step is to extract useful information from these tracks. There is a plethora of different tools that allow the estimation of a variety of variables from the SMT data. Available methods can be classified into three general categories: (a) residence time analysis to extract binding times for stationary molecules, (b) fitting of diffusion coefficient distributions to identify various types of motion, and (c) kinetic models that determine different states of motion occupied by the molecule under investigation. The latter group of methods includes sophisticated machine learning techniques.

MatTrack [38,42] includes the analysis of residence times as well as the fitting of diffusion coefficients. For those that prefer to work in the R programming environment, Sojourner offers similar tools [48].

Bound2Learn is a promising new approach that incorporates a machine learning algorithm to classify molecules as bound or diffusing [23]. Gaussian mixture models of the log of the mean speed of the particle are fed into a Random Forest model, which returns the classification of each track segment. It is available in MATLAB (https://github.com/Reyes-LamotheLab/Bound2Learn, accessed on 10 December 2022).

The application of kinetic models to a collection of tracks or track segments offers the possibility of obtaining multiple states with higher fidelity than is possible with curve fitting. Spot-On is one tool that will return the fractions of tracks within each of up to three states [49]. Spot-On can be used either in the web interface, as a standalone application, or in MATLAB (https://spoton.berkeley.edu/SPTGUI/, accessed on 10 December 2022) and works by calculating the distributions of displacements at each available time lag. This ensemble of displacement distributions is then jointly fit to kinetic models with one, two, or three states. 

An early technique that was developed for the estimation of diffusional states from SMT data is vbSPT [50]. Available in MATLAB, vbSPT (http://vbspt.sourceforge.net, accessed on 10 December 2022) uses a variational Bayesian treatment of a Hidden Markov Model (HMM) to obtain a model that can contain any number of states, with each state characterized by a unique diffusion coefficient. In addition to the presence of the state, vbSPT also estimates the transition rates between states. A major limitation of vbSPT is that it does not take into account localization precision, meaning it may be better suited for the analysis of fast-moving states rather than slow chromatin-bound states. 

Another tool used for calculating an unknown number of diffusion states and transitions from SMT data is perturbation Expectation Maximum version 2 or pEMv2 [51]. pEMv2, which is available as MATLAB code (https://github.com/p-koo/pEMv2, accessed on 10 December 2022), is a machine learning framework that utilizes maximum-likelihood estimation and Bayesian information criteria (BIC) to obtain the diffusion parameters for each track. A global model that explains all track diffusion parameters is found using a perturbation expectation maximum technique. In pEMv2, each track is classified as a single state; therefore, it is necessary to split tracks into lengths that minimize the probability of transitions between states.

Diffusional fingerprinting is another machine learning approach for calculating up to four states and transitions from tracking data [52]. Available in Python code (https://github.com/hatzakislab/Diffusional-Fingerprinting, accessed on 10 December 2022), diffusional fingerprinting calculates 17 features of each track, and a logistic regression classifier is used to determine which state each track belongs to. 

A new method uses a finite state approximation of a Dirichlet process mixture model to simultaneously estimate localization error and diffusion coefficients for each track [53]. This technique, named State Array, is available in Python (https://github.com/alecheckert/saspt, accessed on 10 December 2022) and has the potential to estimate an unlimited number of underlying states within the SMT data, but it cannot estimate transitions between states. 

Simulations are often used in SMT to validate analysis methods or test hypotheses arising from parameters extracted from SMT data. For example, Koo and Mochrie simulated populations of molecules with various diffusion characteristics to show that pEMv2 successfully classified the tracks [51]. Presman and colleagues used simulated tracks to evaluate the ability of curve fitting of survival time distribution (i.e., cumulative distribution function CDF) and residence time distribution (i.e., probability density function, PDF) to extract the correct binding parameters [22]. Kapadia and colleagues simulated the binding of polymerase to replisomes. Comparison of their simulations to experimental SMT measurements indicated that after dissociation from a replisome, the polymerase molecule is unlikely to bind to nearby replisomes [14].

### 2.4. Yeast as a Model for SMT: Pros and Cons

Yeast provides many advantages for SMT. Yeast has a small genome; therefore, many genes are presented only as a single copy, which simplifies genetic manipulations, such as gene tagging with fluorescence markers, controllable protein degradation, introduction of specific mutations in specific genes, the ability to use mutants that will delay specific cell cycle stages, the availability of conditional alleles for numerous proteins to determine their functional impact, and the usage of synthetic genetic systems. Yeast has practically no silent DNA, and all regulatory pathways in yeast are simple and easy to manipulate. Yeast is tolerant to xenobiotic DNA; therefore, yeast is a simplified model for in vivo SMT of the proteins of higher eukaryotes. 

However, conclusions from yeast may not be completely extrapolated to higher eukaryotes. As discussed below, we already know from SMT that the initiation of transcription and perhaps some other cellular processes are partially different in yeast and mammalian cells. This might be predicted for gene expression, as yeast has a compact genome with practically no satellite DNA and no repetitive sequences. Most yeast genes are constitutively expressed, and there are fewer histone variants and histone modifications in yeast than in multicellular eukaryotes. Furthermore, yeast has a low level of DNA cytosine modifications and lacks enhancers. In general, there are fewer gene copies for yeast cellular proteins, which may be indicative of a simplified genetic regulation for many cellular processes.

In addition, there are certain technical disadvantages to using yeast for SMT. The yeast nucleus is much more mobile than in mammalian systems, and this may impose technical problems on long-term observations. In addition, due to its small size, the presence of a cell wall, and the accumulation of autofluorescent biological by-products in a sizeable vacuole, yeast cells have high autofluorescence and light scattering. This results in a reduced SNR and, thus, forces the usage of higher laser power and, consequently, reduces the length of observation time and the number of usable tracks. The small size of yeast cells also affects the size of intracellular protein pools and limits protein traveling distance, thus affecting the specifics of protein kinetics.

## 3. What Did We Learn from SMT in Yeast and Higher Eukaryotes?

In this section, we summarize the current state of the field. Many SMT experiments were performed in insect or mammalian cells; however, our goal is not to give a comprehensive coverage of all SMT research, but to use only a few relevant publications to highlight the overall meaning of the yeast data. Thus, some valuable research performed in higher eukaryotes was left unmentioned in this review. For further reading, we recommend recent reviews [54,55,56,57,58].

Please note, most of the research discussed below is based on thorough studies of transcription. Unfortunately, there are very few SMT studies beyond this field. However, studies of transcription point to a general behavior that should be relevant in some way to other protein complexes, processive or non-processive. All cellular proteins must search and find specific locations in the cell where molecular complexes assemble and function; dynamics of all protein complexes must be optimized; we may expect intricate regulation of complexes that incorporate subunits with different scales of dynamics; mechanisms must exist for dynamic complexes to regulate stepwise continuous processes. Findings in transcription provide a roadmap for interrogating other protein functions by SMT. However, it is expected that the expansion of SMT to other cellular processes will reveal new ways of molecule mobility and dynamics in live cells. 

### 3.1. Kinetics of Protein Complexes

#### 3.1.1. Dynamics of Regulatory Complex Assembly

By ChIP, the main events of transcription activation were defined as orderly and sequential: (1) a pioneer factor binds to regulatory sites covered by nucleosomes, (2) chromatin remodelers are recruited and clear the sites from nucleosomes, making them accessible to Mediator complex and multiple General Transcription Factors (GTFs), (3) GTFs bind to the promoter which facilitates RNA Pol II binding, (4) RNA Pol II binds to the promoter, (5) RNA Pol II recruits more GTFs to form the PIC (Preinitiation Complex). RNA Pol II then escapes from the PIC to transcribe the open reading frame (ORF) (Figure 4A). 

How will transcription activation look if observed by SMT? In yeast, SMT was applied to the components of the PIC [16] and replisome [14]. For PIC formation and dynamics, SMT was performed for the whole population of the TF molecules in the yeast nucleus; thus, parameter estimates reflect behaviors averaged for the same TF binding to multiple housekeeping promoters [16]. The authors demonstrated that PIC assembly is hierarchical: Mediator and General Transcription Factors (GTFs) TFIID, TBP, and TFIIA are necessary for the subsequent assembly of RNA Pol II and other GTFs at promoters. Smaller GTFs may bind to larger TFs before they bind to promoters. Interestingly, components of PIC vary in dynamics (τ in the range of 2–10 s). These insights were confirmed and expanded in in vitro experiments for the binding of GTFs and RNA Pol II to immobilized synthetic DNA segments in yeast nuclear extracts in a molecular environment approximated to that of the yeast cell [59]. The authors were able to separate the binding of PIC to an upstream activating sequence (UAS) from binding to the core promotor, by comparing DNA fragments containing either UAS and the core promoter or only the core promoter. They observed that the individual PIC components may arrive at the UAS of the synthetic promoter sequence in no particular order, as single molecules or preassembled into subcomplexes. Depletion of dNTPs and ATP from the extracts prevented promoter escape of Pol II, and this allowed the authors to slow down and study PIC assembly on the TATA box, which is normally short-lived and difficult to observe. Results from in vitro systems do not always completely correspond to kinetics observed in vivo. If promoter escape was prevented in in vitro experiments, PIC remained assembled on immobilized DNA segments for as long as minutes, much longer than the shortest DNA-bound PIC components, TFIIB, TFIIE, and TFIIF, which had average residence times (τ) of 2 s in live yeast cells [16]. However, as discussed below in more detail, fast kinetics of some replisome components do not lead to disassembly of the whole replisome complex [23]. Therefore, PIC, like the replisome, also may survive in vivo longer than its components that are bound for the shortest time. 

A priori, mammalian and yeast cells might display a difference in PIC lifetime and component dynamics as yeast genes are shorter in length and require less time to be transcribed; the whole yeast genome, as opposed to higher eukaryotes, is much more active transcriptionally, and there are much fewer stalled RNA Pol II complexes in yeast. Indeed, dynamics of TBP and Pol II measured by FRAP are much faster in yeast than in mammalian cells [6,60,61]. Based on SMT in yeast cells, the residence times for Rpb1 (the largest RNA Pol II subunit) and TBP were difficult to estimate because they were close to the survival time of the histones used for the photobleaching control, but they may be on the scale of 20–30 s [16]. In human cells, the residence time estimates are shorter: 7.2 s for the largest subunit of Pol II (POLR2A, or Rpb1) and 9.6 s for TBP [62]. Mediator has a residence time of 3 s in yeast [16] and 18 s in human cells [62]. Therefore, more SMT studies are required to compare the dynamics of transcription in mammalian and yeast systems. However, we may conclude that branched alternative paths of building a molecular complex exist both in higher eukaryotes and in yeast, and subunits of the same molecular complex may display different dynamics (Figure 4B).

#### 3.1.2. Dynamics of Processive Complexes

What dynamics are displayed by RNA Pol II complex engaged in the processive elongation ? Is it stable during a single elongation cycle? What are the dynamics of its components through the single elongation cycle? 

Direct measurements of the progress of the processive enzymes are beyond the resolution of currently available imaging technology. To detect any directional movement, long tracks with fast temporal sampling are required; however, photobleaching and loss of focus limit either the rate of sampling or the number of timepoints; fast sampling will lead to short tracks, and a low sampling rate will create problems in linking the correct molecules in sequential frames. Moreover, while detecting binding, we cannot distinguish between true binding or site sampling due to the inadequate spatial resolution of the typical SMT experiments. As an example, if the average speed of the replication fork is 1.6 kbp/min, and the size of a half-replicon replicated by a single replisome is 18.5 kb, then assuming for simplicity that the DNA template is a straight filament, and assuming a typical SMT resolution of 30–40 nm, it will take ~3.5 s for the replisome progress to be detected. However, if, as expected, replicated DNA is in a loop or a more complex spatial configuration, the progress of the replisome will be completely undetectable with the current spatial resolution of the SMT technique. Similar considerations apply to RNA Pol II during elongation if the average elongation rate is 3.9 kbp/min [13]. Therefore, current technology allows us only measurements of the residence and search time.

It is hard to estimate the residence time of the RNA Pol II elongation complex in vivo by SMT in ensemble studies. A residence time of 23 s was reported for the biggest subunit of RNA Pol II [16], but this value is averaged for PIC and the elongation complex. After depletion of kinase TFIIH, which is required for promoter escape, the residence time of Pol II was shortened to 10 s. This value characterizes not only a population of Pol II in the PIC, but also the residual population already present in elongation complexes that were formed before TFIIH depletion. Indirect information on the stability of the elongation complex was provided by SMT of the FACT complex, a histone chaperone, which associates with DNA after the initiation of transcription [63]. The residence time of Spt16 (a subunit of FACT) was within the range of Pol II residence time (τ = 20–25 s). As SPT16 acts only at the post-initiation stages of transcription, the similarity of RNA Pol II and Spt16 residence times indicates that elongation is a dominant component in residence time estimates of RNA Pol II by SMT. Adopting an elongation rate of 65 bp/s [13], τ = 20–25 s should be sufficient for the transcription of the average yeast ORF of ~1600 bp by a single elongation complex. The transcription elongation factor SPT5, a component of the DSIF complex, stabilizes RNA Pol II at the promoter-proximal regions [64], but also promotes the release of stalled elongation complexes [65]. From in vitro measurements conducted in yeast extracts, the transcription elongation subcomplex Spt4/5 was associated with synthetic immobilized reporter DNA and the Rpb1 subunit of RNA Pol II for the whole duration of the transcription event [65]. However, the residence time of SPT5 was reported to be 5.5 s in vivo in mammalian cells, which is much shorter than the average duration of a typical transcription event [62]. This indicates that within the same elongation complex, different SPT5 molecules are interchangeable and, furthermore, that subunits of a single processive transcription elongation complex may vary in dynamics, similarly to PIC subunits in the experiments discussed above. This raises the question of how a molecular complex can steadily proceed with the elongation of nascent mRNA molecules while its components pop in and out of it. We may learn something about this mechanism if we compare complex elongation dynamics with the dynamics of another processive molecular complex, such as the replisome.

Kapadia et al. explored the intranuclear dynamics of DNA polymerases at the S stage of the cell cycle when polymerases are predominantly assembled into replisomes [14]. This molecular machine consists of three different polymerases, two of which, Pol ε and Pol δ, move in opposite directions as they synthesize DNA in leading and lagging strands, respectively. Although two polymerases, Pol ε and Pol δ, move in opposite directions, the whole replisome moves in one direction. Thus, Pol δ must be periodically separated from DNA to move to a different area. However, unidirectional processivity is best compatible with the stable association of Pol ε and Pol δ throughout the lifetime of a single replisome, which is confirmed by SMT. Pol ε and Pol δ residence times were measured to be extremely long (~5 min) in vivo, with computer simulations supporting the measurements, indicating that Pol ε and Pol δ assembled into a processive complex and stayed together until the end of the processive activity. 

The third polymerase, Pol α, is involved in RNA primer synthesis and extension, and therefore, must bind to DNA cyclically. One may expect that Pol α falls off the template after the synthesis of each short primer. However, the long residence time of Pol α (64 s) indicated that the same molecule may be involved in several cycles of priming activity; although, in the same replisome, individual molecules of Pol α are interchangeable. 

Thus, processive complexes appear to be more stable than regulatory complexes. At the same time, within the same complex, different subunits still may have different kinetics. Biologically, this means that the complex must hold together somehow while proceeding along the DNA, even when some components are temporarily disassociated from DNA. One possibility for such behavior is tethering the processive complex to the DNA, while some of the subunits temporarily separate from the DNA template but retain a connection with the replisome. Recently, a mechanism of tethered diffusion was described for kinesin in in vitro experiments. The authors observed the mobility of fluorescently labeled kinesin along microtubules attached to the coverslip. They discovered that the kinesin cofactor MAP7 was tethering kinesin to the microtubule and allowed it to sidestep obstacles by temporarily dissociating from the microtubule but still retaining connection with the same microtubule via MAP7 [66]. Tethered diffusion for kinesins or other processive complexes has not yet been observed in vivo.

### 3.2. Fast Kinetics of TFs Translated into Slow Kinetics of Transcription Bursts

Regulatory complexes display fast dynamics; however, they may regulate processes with much slower dynamics. A good example is transcription, which coincidentally is a process extensively studied for molecule dynamics. The dynamics of individual TS observed in live cells by stem-loop labeling of nascent mRNA have demonstrated that for many genes in different organisms, transcription occurs not continuously but in periods of transcriptional activity (bursts) interspersed with periods of transcriptional silence (see [54] for review). During a burst, multiple polymerases may be associated with an ORF, and many mRNA molecules may be produced, and a burst may last for many minutes or even hours. How are slow kinetics of the bursts driven by much faster dynamics of TFs and the PIC? The same question may be applied to other less-studied cellular processes (Figure 4C). 

#### 3.2.1. Short Binding Events May Lead to Long Bursts of Transcription

Can very short binding events lead to the initiation of transcription? One may suggest that, alternatively, initiation of transcription might be driven by a very small fraction of TF molecules with a very long residence time, while the population with shorter binding events detected by FRAP and SMT is futile. However, this hypothesis has been disproven; links between short binding events and transcription bursts were established in mammalian cells and yeast [4,12,13,67,68,69,70]. For the yeast *CUP1* locus and transcription activator Ace1p, the immobile fraction of Ace1p revealed by FRAP was not sufficient for the level of *CUP1* activation observed in a population of yeast cells [4].

Subsequent studies further revealed and quantified the role of fast TF kinetics in transcription bursts. Direct, simultaneous observation of TF binding and transcription from the same promoter is technically challenging, thus many publications rely on combined approaches: TF *k_on_* and *k_off_* rates and binding rates are estimated from SMT data; parameters of transcription are estimated by modeling nascent and mature mRNA distributions in fixed cells, and dynamics of nascent mRNA accumulation in live cells. Bursts are characterized by the activity (the ratio of burst time to total time of observation), burst and inter-burst duration, burst amplitude (burst intensity, or polymerase load, calculated as an average number of polymerases or nascent mRNA per TS at any given time point), burst size (total number of mRNA produced per burst), and burst frequency (Figure 4D). The ON and OFF states of the gene are defined as the gene in the state of active transcription and gene in the inactive state, respectively. Typically, the gene ON state is considered different from burst duration; the gene ON state is a state in which transcription may be initiated, while a burst is a state in which mRNA can be detected at a TS. In the ON state, the gene becomes competent for transcription, although this state does not always lead to the production of an mRNA transcript. Furthermore, there may be a time gap between switching the gene to the ON state and the appearance of the first mRNA, and the last mRNA may still be visible at TS even if the gene switched to the OFF state, and initiation of new mRNAs had already stopped. 

It was demonstrated for the yeast *CUP1* locus that its *k_ON_* is driven by *k_on_* of transcriptional activator Ace1p [12]. The authors tested the hypothesis that the short binding events of TF lead to gene bursts, and consequently, estimates of TF *k_on_* and *k_off_* from SMT accurately represent gene *k_ON_* and *k_OFF_* rates. The authors modeled the distribution of mature mRNAs from a reporter driven by the *CUP1* promoter, assuming that the gene may only be either in the ON or OFF state (“random telegraph” model), and assuming that the switching of the *CUP1* promoter to the ON state is triggered by binding of at least one Ace1p molecule. They inputted two parameters into this model: mRNA degradation rates, which were estimated separately, and the *k_off_* of Ace1p as the gene’s *k_OFF_*. The gene’s *k_ON_*, which was obtained from Monte-Carlo simulations during the fitting process, was very similar to the TF *k_on_* calculated from the SMT data. If the TF *k_on_* and *k_off_* would not be related to gene *k_ON_* and *k_OFF_*, then the returned *k_ON_* value would be different from TF *k_on_*. This indicated that the short binding events of Ace1p are the drivers switching the promoter to the ON state. 

This conclusion was further supported by direct observation of TF binding and subsequent transcription bursts at the same promoter. Two groups took advantage of the uncommon technology of Orbital Tracking in mammalian cells [70] and yeast [13]. In yeast, multiple sequential binding events of Gal4 at the same TS of *GAL10* were observed as short spikes of Halo Tag/JF_646_ fluorescence (τ = 35 s); they were followed by longer spikes in PP7-GFP fluorescence (τ = 152 s), which indicated that Gal4 binding precedes RNA appearance at the TS. Therefore, Gal4, like Ace1p, switches the promoter to an ON state.

Moreover, Gal4 is present at the promoter when transcription starts, as an overlap of 14 s was detected between the residence of Gal4 at *GAL10* TS and the appearance of nascent mRNA. This indicates that TF in yeast may be required for the gene to finalize PIC formation and RNA Pol II promoter escape. Furthermore, when a TF leaves the promoter, initiation of transcription stops, as demonstrated in a synthetic system in yeast where the binding of synthetic TF to the synthetic reporter was controlled by a photoactivable domain. Without photoactivation by blue light (405 nm), the TF was deactivated and left the promoter within a minute, and the initiation of transcription was abolished [69].

Taken together, observations from these three papers [12,13,69] lead to the following conclusions: (a) short binding events of transcription activator can switch the gene into ON state and (b) the presence of a transcription activator is necessary for the burst to continue; the departure of the activator from the promoter switches the gene into an OFF state. 

#### 3.2.2. Yeast and Mammalian TFs Play Different Roles in Initiation of Transcription

The relationship between TF binding and transcription bursts in mammalian cells is more complex than in yeast, and it cannot be described by a simple two-step ON/OFF random telegraph model [70,71]. For the Orbital Tracking of Glucocorticoid Receptor (GR) at the synthetic MMTV promoter-based reporter, nascent mRNAs were detected by the stem-loop approach, while GR was detected by a HaloTag [70]. Similar to the Gal4 parameters observed by Orbital Tracking in yeast cells, GR had a residence time of 56 s, while the nascent mRNA had a residence time of 156 s. The shift between the two peaks of fluorescence intensities was 182 s. The authors did not state whether there was any overlap between the residence of GR and nascent mRNA residence. However, applying the same calculations as in Donovan et al. [13] to the GR data, we find no overlap between the bound GR and nascent mRNA at TS; conversely, there is a gap of ~76 s. Thus, GR leaves before transcription starts, and therefore, the continuous presence of GR is not required for the ON state of the promoter. 

These conclusions were confirmed in another mammalian system [71]. The authors assayed the effects of variable binding affinities and the molecular concentration of synthetic Transcription Activator Like Effectors (TALE) on transcription bursts of the synthetic reporter. The synthetic reporter contained a single tetO site in its promoter, to which different TALE-TFs could bind with different affinities that were estimated from the binding of each TALE-TF to the multiple consensus sites naturally present in the genome. The transcription from a synthetic reporter was detected by the stem-loop RNA PP7 approach and by RNA smFISH. Based on observations and data modeling, the $k_{on}^{*}$ of the TF-regulated burst frequency did not influence burst duration or burst size. These data could not be explained by a widely applied random telegraph model, which assumes that the gene may only be in the ON or OFF state. The relationship between TF binding and transcription in these experiments was best explained by an extended three-state model, where the TF only acted to prime the promoter for transcription to occur, but its presence was not necessary for the subsequent steps of initiation. 

Therefore, higher eukaryotes are different from yeast cells. In higher eukaryotes, TFs do not need to be present for the whole process of initiation. An interesting hypothesis links this feature with the evolution of enhancers [71]. Enhancers supply extra target motifs for TFs. However, enhancers come only in transient contact with core promoters. Therefore, TFs bound to enhancers also encounter core promoters only transiently; thus, transcription may be more efficient if initiation is finalized in the absence of TFs.

### 3.3. Search for the Specific Binding Sites

Regulatory factors must search for their target sites in a complex nuclear landscape, which may obstruct or facilitate their search process due to changes in local protein concentration and the presence of sizeable nuclear bodies. Searching proteins must compete with multiple factors binding to the same targets, specifically or nonspecifically. This applies to any protein and any target site, such as DNA, RNA, protein aggregates, nuclear bodies, or cytoskeletal components. Thus far, the search process has been studied only for proteins binding to DNA, partially by convenience, as this type of binding occurs on a well-characterized target, DNA, with known consensus motifs, and furthermore, the binding of proteins to DNA can be reproduced in vitro to some extent. Mechanisms of nonspecific binding were assayed in bacteria, yeast, and mammalian cells (see review [57]). 

For most DNA-binding regulatory factors, multiple binding sites with different affinities are present in the genome, some of which match the consensus sequence. The latter are typically located in regulatory elements and perform specific regulatory functions. The others with different levels of mismatch to the consensus, are located randomly and they do not play any regulatory role and have no specific function. In eukaryotes, it is unclear how many of the putative binding sites are available for binding, as access to many of those may be obstructed by nucleosomes. 

#### 3.3.1. Protein Interactions with DNA during the Search Process

According to the common observations, the search for binding sites by TFs occurs by the sampling of DNA, i.e., by transient nonspecific binding alternated with diffusion. In yeast, the existence of nonspecific binding was first demonstrated by FRAP for the TF Ace1p. Ace1p requires Cu^2+^ for binding to a few specific sites in the promoters of only five genes; therefore, any binding detected in the absence of Cu^2+^ is nonspecific. In the nucleoplasm, Ace1p displayed much slower mobility than the free GFP molecules. Modeling of this process indicated that Ace1p moved by effective diffusion, i.e., it spent more time bound than diffusing [4]. Later, in SMT experiments, the nonspecific binding of Ace1p was quantified and compared to the nonspecific binding of heat-shock factor HSF and histone H3. The similarity of the residence time for several proteins with vastly different functions pointed to a common mechanism(s) of nonspecific interactions with DNA for different proteins [11]. 

What is known about the modes of protein mobility during the search for specific target sites? Most of the information comes from in vitro studies. TFs are expected to interact with DNA by facilitated diffusion, consisting of 3D translocation between DNA segments and 1D hopping and sliding along the DNA (reviewed in [72]) (Figure 4E). The plausibility of one-dimensional (1D) diffusion was demonstrated in vitro on stretched fragments of DNA [73]. The observation of 1D diffusion in vivo has thus far been impossible, much less distinguishing hopping from sliding, due to insufficient spatial resolution. Thus, it is still unclear how often 1D diffusion occurs in vivo. Modeling of SMT data indicates that 1D diffusion may happen in bacterial cells [74] and mammalian cells [75]. For the bacterial TetR microinjected into mammalian cells and binding to quasi-consensus sites, facilitated diffusion is probable, and the protein may travel in stretches in the range of 250–750 bp by 1D diffusion [75]. 

A priori, sliding along the DNA in eukaryotic cells might be affected by histones. However, as demonstrated in vitro, 1D diffusion may occur even on a target covered by nucleosomes [76]. Supposedly, 1D diffusion may consist of sliding along the naked DNA and hopping over the obstacles encountered along the DNA, such as nucleosomes, TFs, or other proteins. For the yeast chromatin remodeler SWR1, it has been shown that hopping over obstacles is more likely than sliding because sliding of SWR1 on stretched DNA in vitro is obstructed by bound proteins [77], and there would be even more obstructions in vivo.

Bacterial regulatory proteins spend most of the time bound to DNA, but the regulatory proteins in eukaryotes spend more time unbound. This may result from multiple factors, including the rarity of naked DNA and the much bigger size of the eukaryotic genome meaning more sites to search. As observed by SMT, in *E. coli*, LacI spends ~90% of the time bound and gliding along the DNA and only 10% in the diffusing state [74]. In mammalian cells, conversely, LacI, as well as bacterial repressor TetR, spend 80% of the time diffusing and only 20% of the time bound to the numerous quasi-consensus sites present in mammalian DNA [75]. The authors concluded that in mammalian cells, binding for TetR and LacI is highly inefficient, and their search process is limited not by diffusion, as in bacteria, but by binding to non-specific sites. This indicates that the nuclear landscape and chromatin dictate the sampling mode.

Interestingly, some proteins do not search for DNA but instead search for the proteins associated with DNA, or alternatively, they may bind to DNA only if bound to another protein. Proteins that do not bind to DNA on their own should be incapable of 1D diffusion. Indeed, bacterial RNA polymerase, as opposed to bacterial regulatory protein LacI, is incapable of 1D diffusion; it searches for a binding site via 3D diffusion, as observed in vitro [78]. In yeast cells, RNA Pol II and small GTFs do not associate with DNA if they are not in a complex with Mediator and TFIID, and thus, it may be assumed that they also search for targets only by 3D diffusion [16].

#### 3.3.2. Nuclear Space Sampling during the Search Process in Eukaryotic Nucleus

SMT in yeast showed that the space exploration by the GTFs may be constrained to areas smaller than the volume of the whole yeast nucleus [16]. In fast-acquisition SMT for Pol II, Mediator, TATA Binding Protein (TBP), and GTFs, two populations were identified—diffusing and bound. For small GTFs, the bound population could be further separated into sub-diffusive and chromatin-bound states. For Pol II, Med14 (a subunit of Mediator), TAF1 (a component of TFIID), and their parameters (D = 0.6 µm^2^/s, and *R_c_* = 0.4 µm) indicate that they sample areas smaller than the whole nucleus. Small GTFs (TBP, TFIIA, TFIIB, and TFIIE) in diffusing state ( D > 2 µm^2^/s) display *R_c_* = 0.75 µm, that corresponds to the radius of the nucleus, indicating that they may explore the whole nucleus. In sub-diffusive states with D = 0.4–0.6 µm^2^/s, small GTFs are confined to smaller areas with *R_c_* = 0.4 µm, similar in size to those of Mediator and TFIID, indicating that this subfraction of small GTFs associates with larger GTFs. Indeed, the sub-diffusive state of small GTFs disappears if Mediator, RNA Pol II, and TFIID are depleted. Thus, smaller GTFs may search for target sites as individual molecules (diffusive sate) or in association with bigger GTFs (sub-diffusive state). The chromatin-bound state of PIC components and histone H2B displayed tight spatial confinement (*R_c_* = 0.13 µm). 

Different modes of space exploration were described by SMT in more detail for the regulatory proteins in mammalian cells. Two proteins overexpressed in human osteosarcoma U2OS cells, the general transcriptional activator, c-Myc, and the transcription elongation factor P-TEFb, sampled the space in a different way. Modeling of the distribution of the angles between consecutive translocations indicated that c-Myc is a non-compact explorer, while P-TEFb is a compact explorer. Non-compact explorers search the space by isotropic diffusion, but compact explorers search the space by anomalous diffusion, with obstructed mobility [79]. The authors predicted that the compact searchers are the proteins, such as P-TEFb, that contain domains facilitating their aggregation and accumulation in temporary clusters, while proteins without such domains, such as c-Myc, perform non-compact searches (Figure 4F). The space explored by P-TEFb was best described as a 2D fractal lattice. P-TEFb phosphorylates the Carboxy-Terminal Domain (CTD) of RNA polymerase II. The authors suggested that P-TEFb moves along the putative meshwork formed by CTD. Interestingly, although c-Myc moves by isotropic diffusion, a subfraction of its population is confined to small 300 nm areas. A third mode of search, Anisotropic Diffusion through transient Trapping in Zones (ADTZ), was later described for CTCF, which is the factor demarcating spatial domains in chromatin [80]. CTCF sequentially explored small areas, i.e., it was transiently trapped in small retention zones, which most probably corresponded to the microscopically observed CTCF clusters. This behavior depended on the CTCF domain RBR_i_. Deletion of this domain abolished the trapping of CTCF and led to exploration of the whole nucleus. Thus, different proteins may rely on different searching strategies in the same molecular environment. Moreover, the same protein may behave differently in different cell types. In a separate study, in non-cancerous HBEC3-KT cells, a greater fraction of c-Myc was specifically bound as compared to c-Myc in the human osteosarcoma U2OS cells (98% versus 46%) [62].

### 3.4. Regulation of Promoter Occupancy 

High time-averaged promoter occupancy by transcription activators may occur due to long residence (low *k_off_* of TF) or frequent binding (high *k_on_* of TF)—which is more important for gene regulation? 

#### 3.4.1. Factors Affecting Frequency of Protein Binding to Target Sites

An increase in the binding frequency of a TF may improve the transcription rate, as demonstrated in [67,68,69,71] and other studies. The binding frequency depends on the availability and accessibility of the target motifs and availability of the TF. To find between one and six binding sites of ~10 bp length in a specific promoter, DNA-binding proteins must sample a genome of 12 Mbp in a volume of 4.2 μm^3^ in a diploid yeast nucleus, or a genome of 3500 Mbp, in a volume of 500 μm^3^ in a mammalian nucleus. From SMT experiments, one may estimate the search time (τ*_search_*) for a specific TF, the number of TF molecules, and the number of binding sites. From these estimates, one may calculate how long it may take for the single protein to find a single target site in a cell of an appropriate volume. It will take 30 min for a single LacI in a very small *E. coli* cell [74], 5 hours for a single Mbp1 in yeast [81], 31 days for a single Sox2 molecule in mammalian cells [39], or 35 days for bacterial TetR lost in an unfamiliar mammalian cell landscape [75]. Therefore, cells must compensate for this unacceptably slow search time by either increasing the number of target sites available for binding or by increasing the molecular concentration of the protein.

Amplification of target motifs should increase the probability of binding of the TF and thus increase the time-averaged occupancy of the promoter by TF. Many promoters contain multiple binding sites for the same TF, and it was shown that amplification of target sites in a promoter increases the transcriptional output, probably due to higher occupancy of the promoter by TF. For instance, *GAL3* in yeast contains only one binding site for Gal4, but *GAL10* has four binding sites, and transcription from *GAL10* is much more efficient [13]. However, there may be a limit on target motif clustering, as observed in yeast on synthetic arrays of tetO sites with varying lengths. Occupancy of these arrays increased with an increase in the number of binding sites, but eventually reached a plateau. All arrays with over 67 repeats had the same number of occupied sites. If the same number of repeats was not arranged in tandem, but dispersed in the genome, they were occupied with much higher efficiency [82]. 

Different modes of search may have implications for the regulation of gene expression. Non-compact search is affected by the size of the target of the search and does not dependent on travel distance; compact search depends on the travel distance and does not depend on the size of the target. For example, from modeling of SMT data in mammalian cells, the estimated time needed to find a 10 nm target located at 250 nm distance was 506 s for c-Myc (non-compact searcher) and 7.4 s for P-TEFb (compact searcher), but if the target was located 5 μm away the search time was approximately the same (525 s) for c-Myc but substantially longer (64.6 s) for P-TEFb [79]. A non-compact searcher has an equal probability of binding to all loci because it leaves a lot of targets unsampled. Compact searchers, on the other hand, have a higher probability of binding to adjacent loci by performing a redundant search (Figure 4F). Therefore, compact searchers should have a higher probability of regulating adjacent loci. Another predicted consequence of the compact search is that the same TF may repeatedly visit the same target site with a high probability. Therefore, the frequency of binding and time-averaged occupancy may be improved for a compact searcher if target motifs are clustered and the search for those is confined to a small area. In other words, confinement of the protein to a small area is equivalent to increasing its local concentration.

Several papers provide direct or indirect evidence that cells may maintain optimal local concentrations of the TFs. Indeed, although an increase in the molecular concentration of the TF reduces the search time and leads to an increase in transcriptional output [67,68,69,71], the total protein concentration in the nucleus obviously cannot be increased indefinitely. Alternatively, the local concentration of protein may be increased. Visible evidence for variability in local concentrations includes dynamic small clusters or hubs formed by proteins in the nuclei of mammalian cells. RNA Pol II forms dynamic clusters or hubs with a lifetime on the scale of seconds [83]. Clusters of RNA Pol II can associate with clusters of Mediator [84] and with clusters of the enhancer protein SOX2 [10]. SMT reveals that SOX2 moves between clusters by diffusion and persists in clusters mostly in the chromatin-bound state [10]. Those aggregates are large enough to be detected microscopically. It is not always clear whether all such clusters are functional or if some of them are just a by-product of artificial protein overexpression or the propensity of some protein domains to form temporary aggregates.

Not much is known yet about the mechanisms controlling variability in protein concentration and the formation of protein clusters. Proteins may be sequestered by binding to bigger protein complexes, as demonstrated in yeast, where Mediator and TFIID associate with smaller GTFs and sequester them in small subnuclear areas [16]. Regulatory proteins may assemble into micro-clusters on tandem repeats of the target sites at the same promoter or enhancer, as observed in vitro. For instance, on synthetic promoters in yeast extracts, multiple molecules of RNA Pol II may be recruited to UAS by transcriptional activators bound to five tandem repeats of the Gal4 target site. The authors infer that in live cells, multiple binding sites for transcriptional activators may create a depot for the TF and thus facilitate the bursting of transcription [59]. Mammalian cells have microsatellite DNA with repetitive repeats that may serve as seeds for the hub formation, as described for the oncogene EWS:FLI1, which aggregates on microsatellite GGAA repeats, and this aggregation promotes transcription [85]. 

Recently, it was discovered that proteins may assemble into temporary clusters via liquid–liquid phase separation (LLPS), facilitated by Intrinsically Disordered Domains (IDR), which include the subcategory of Low-Complexity Domains (LDRs) (see review [86]). Supposedly, intracellular hubs, which are smaller than LLPS clusters, and thus non-detectable by microscopy, may also arise through LDR interactions. Micro-hubs may be facilitated by regulatory factors promoting aggregation at specific sites via LDR domains. The domain CTD of RNA Polymerase II is an LDR that establishes contacts between multiple Pol II molecules and facilitates its aggregation into small droplets in vitro. In yeast, CTD controls burst size and frequency by facilitating the binding of multiple RNA Pol II molecules to the same paused PIC, and by slowing down the release of RNA Pol II [87]. The authors demonstrated that overexpressed LDRs may force LDR-containing proteins to aggregate, but no microscopically visible hubs of RNA Pol II were reported in this study. The authors report that in genomes, where genes are located sparsely, RNA Pol II tends to have a longer CTD, presumably to facilitate contacts between multiple sparsely bound RNA Pol II molecules. In *Drosophila melanogaster* embryos, there is a gradient in the concentration of transcription factor Bicoid. Factor Zelda compensates for the drop in Bicoid concentration and maintains the Bicoid ON rate by promoting hubs of Bicoid on specific target sites [88]. However, aggregates may become too large and impair transcription, as observed in mammalian cells with overexpression of the domain LCD of the oncogene EWS:FLI1. Overexpressed LCD sequesters the wild-type protein, which leads to a reduction in gene transcription in the vicinity of the aggregate [85]. There may be an optimal level of protein clustering in hubs or depots, above or below which transcription may be impaired.

#### 3.4.2. Factors Affecting Protein Residence Time on Target Sites

An increase in the residence time of transcription factors in mammalian cells may improve transcription [9,89,90]. In yeast, in vitro estimates of the affinity of Gal4 to different UAS sequences were compared with in vivo burst parameters for the variants of the *GAL3* promoter containing appropriate UAS sequences [13]. The lower affinity of Gal4 correlated with lower transcription output, and higher affinity supported larger burst sizes. However, there was a limit to the improvement of transcription, which was equal to the level of affinity of Gal4 to wild-type specific motifs. Further increase in the affinity of Gal4 did not improve the transcription. Without performing SMT, it is hard to explain what exactly happens at the promoter in live cells. Maybe the long residence time of Gal4 on high-affinity sites simply does not lead to higher efficiency of transcription. Alternatively, in vivo Gal4 residence on those sites may not be as high as in vitro due to yet unknown factors. 

Similarly, a reduction in the binding affinity at active transcription sites was observed in a synthetic system in yeast [82]. The authors introduced into yeast a synthetic reporter gene with a promoter containing seven tetO binding sites for a synthetic transcriptional activator. Arrays of “decoy” tetO sites inserted at other sites were competing with the promoter for the same TF. The authors measured the tetO site occupancy by ChIP and measured reporter expression level by the accumulation of a fluorescent reporter protein. The modeling of occupancy and expression demonstrated that the residency of TF was lower on active promoters compared to the decoy sites. Thus, *k_off_* of TF on the same consensus sequences may be modulated.

The residence time of a single molecule at the promoter is determined by its affinity to the binding site; the lower the *k_off_* rate, the longer the molecule will be bound to the site. However, external factors besides affinity may affect the residence time of TFs. Mostly, those factors are yet unknown, except for the chromatin remodelers. FRAP data for the nucleoplasmic mobility of Ace1p indicate that chromatin remodeler RSC actively removes Ace1p stuck at nonspecific sites [5]. Thus, the residence of nonspecific sites is not solely determined by the binding properties of the TF, but also depends on the specific action of chromatin remodelers.

As observed by SMT, RSC plays two seemingly opposing roles on specific motifs at the *CUP1* promoter: residence time of Ace1p is reduced by RSC, but the occupancy of the promoter (*k_on_*) by Ace1p is increased [12]. This makes sense if RSC remodels the chromatin target in rapid cycles, leading to increased accessibility of the target motif for Ace1p followed by the removal of Ace1p. If RSC function is impaired, Ace1p is no longer efficiently removed, and its residence time increases. At the same time, access of Ace1p to the binding sites is also impaired, which leads to less frequent binding of Ace1p to the promoter, and lower promoter occupancy by Ace1p. Indeed, the authors demonstrated rapid cycling of RSC and fluid chromatin structure at the *CUP1*-activated promoter, indicating rapid changes in the accessibility of the target motifs to Ace1p. RSC depletion leads to moderate changes in *CUP1* transcription output from the cell population; however, this translates to detectable phenotypic effects, such as higher sensitivity to copper and elevated variability in transcription response between individual cells. This indicates that optimal, rather than long, residence time may be beneficial for cell survival.

Kim et al., 2021 carefully studied the dynamics of all yeast remodeler complexes and discovered that, as opposed to RNA Pol II and large subunits of PIC, all chromatin remodelers display very short residence times [17]. Fast dissociation rates of chromatin remodelers depended on their ATPase activity, indicating that they were the consequence of remodeling activity. The authors explained this quick exchange by competition between different types of chromatin remodelers, which may bind simultaneously to the same promoter. Thus, not only RSC but all other remodelers undergo fast cycling, which indicates that not only Ace1p but many other TFs at yeast promoters may undergo remodeler-induced rapid cycling. 

A couple of studies indicate that other proteins, besides chromatin remodelers, may curtail the residence time of TFs at promoters. FRAP data demonstrate that the kinetics of TBP and TAF1 are regulated in yeast by *MOT1*, which encodes TBP-associated ATPase. In the absence of Mot1p, the residence time of TBP and TAF1 on DNA is increased, indicating that Mot1p is responsible for the removal of non-functional PIC and restoration of the available TBP pool [6]. In mammalian cells, overexpression of the oncogene c-Myc reduces the residence time of MED1, a subunit of the mediator complex [62]. 

Alternatively, there appears to be a lower limit on residence time, below which TF binding to the promoter is futile. In mammalian cells, this was discovered in a synthetic system, which is discussed in Section 3.2.2 in more detail. Modeling of the data indicated that there should be a threshold for the productive residence time of TALE-TFs at the synthetic promoter: a binding event above this threshold is productive, i.e., leads to active transcription, but a binding event with shorter residence time is futile [71]. In yeast, there is indirect evidence of a threshold for productive binding events at the promoter of *CUP1*, where for shorter residence times (τ = 2 s), only one out of three binding events of Ace1p are productive, but for longer residence times (τ = 5 s), each binding of Ace1p is productive [12] (Figure 4G). No futile binding events for Gal4 were observed directly in Orbital Tracking experiments at TS of *GAL10* [13]. It is worth noting that the Gal4 binding events in these experiments were longer than those detected in conventional HILO experiments (τ = 35 s). It may be that shorter binding events are under-detected in movies collected with 250 ms time resolution. However, transcripts initiated by non-detected Gal4 binding should still be detectable, as the minimal duration of the burst should be 32 s based on direct measurements reported in the same paper. If “Gal4-less” spikes of mRNA were not observed, this indicates that any unobserved shorter Gal4 binding events are futile and point to a lower limit on the residence time threshold for productive TF binding. 

#### 3.4.3. Long Residence or Frequent Binding?

For some proteins at some promoters, residence time appears to be extremely short, and yet a high occupancy of the promoter is achieved. In yeast, chromatin remodelers RSC and SWI/SNF display high occupancy (30% for RSC and 94% for SWI/SNF) and low residence time (5 s and 4 s, correspondingly), due to a short search time and high frequency of binding. This mode of achieving high time-averaged occupancy is tailored to the mode of action of chromatin remodelers, allowing simultaneous binding of different remodelers to the same promoter and ensuring rapid dynamics in the accessibility of the target motifs in the promoter [17].

In *Drosophila melanogaster* embryos, the residence time of the TF Bicoid (BCD) is extremely short, ~1 s, and thus, promoter occupancy is dominated by *k_on_* and maintained by local enrichment for TFs at specific locations [88] (Figure 4H). The authors argue that certain types of TFs, including TFs involved in development, bind to low-affinity targets; occupancy of these targets is facilitated by local crowding of specific factors in the vicinity of specific sites ensuring these factors outcompete nonspecific factors. In general, the majority of TFs most probably regulate burst frequency by high occupancy due to their high concentration [91].

The short residence time of TFs is crucial for a Transcription factor Activity Gradient (TAG) hypothesis, which predicts that enhancers may generate gradients of local effective concentrations for TFs [92]. The authors speculate that TFs bound to enhancers may be acetylated by regulatory factors that they recruit. Acetylated TFs will have a higher probability of binding to the target motifs in the core promoters. Due to high intracellular concentrations of deacetylases, those TFs will be frequently inactivated when they diffuse from the enhancer. Therefore, the high concentration of acetylated TFs will be maintained only for recently released TFs, and therefore acetylated TFs will reach only promoters in the vicinity of the enhancer. Importantly, this mode of regulation requires constant production and constant release of acetylated TFs. Short residence time permits the fast escape of acetylated TFs and their isotropic diffusion from the enhancer. It is unclear whether this mode of regulation may be applicable to yeast. 

So, what is more important—regulation by frequent binding or by prolonged residence time? Most probably, it is determined by the requirements of a particular system; to confirm that, we need to study multiple genetic models, encompassing housekeeping and inducible genes. Rapid dynamics at promoters may provide more efficient regulation. It has been demonstrated that regulation by the frequency of bursts is more efficient than regulation by burst amplitude, and the former may lead to more stable and higher transcriptional output [93]. Moreover, is it really “the longer the better” in a complex cellular environment, where many unrelated factors compete for the same binding sites, specific or nonspecific? Dynamic binding gives every protein a fair chance to find the specific motif. See [58] for further discussion.

## 4. Conclusions: Hopes for the Future

Single molecule tracking was mostly applied to transcription factors and factors of replication. Many lessons learned from these experiments may be applicable to other protein factors in the cell. In the future, one may anticipate SMT to be applied to factors of translation, molecular motors, etc. We still have a lot to learn about the behavior of individual proteins, and current imaging technology may still deliver a lot of answers. However, whatever we know is limited by spatio-temporal resolution and throughput. 

Currently, we cannot observe the behavior of two or more subunits in the same molecular complex, such as the replisome and the PIC; thus, we cannot observe subunit interactions directly, and we must deduce the behavior of the complex from the dynamics of its components observed individually. We cannot directly observe and quantify the work of processive enzymes due to the spatio-temporal resolution limitations of the current SMT technique. We need to improve our ability to track single molecules in 3D and to track several proteins simultaneously, we need to achieve higher spatio-temporal resolution without loss of the signal due to photobleaching, and to achieve high-throughput data acquisition and high throughput automation of the analysis. 

New and improved technology can help to solve these problems. Suppose we have those new tools. What will we be able to do with this new technology? Eventually, we want to study a cell in its entirety. The big questions are how the coordination of different processes occurs in the nucleus and what role does the microenvironment play? Many things happen at the same location at the same time. How do different proteins go about their business, how do their independent functions fit together, and what prevents unrelated processes from interfering with each other? Following multiple proteins at a time in their local environment provides information about the ensemble work of unrelated processes. We will observe the formation of the complexes and different protein dynamics in different cellular compartments in real time. We will understand how the different steps of gene expression (transcription and translation, post-translational modifications, and delivery to cellular compartments) work together and overlap with replication, recombination, and repair. We want to be able to do in vivo all that we can currently do only in vitro, and more. We started with individual proteins, but eventually, we want to know and understand how all the separate pieces fit together in a living cell. 

## Figures and Tables

**Figure 1 ijms-23-15895-f001:**
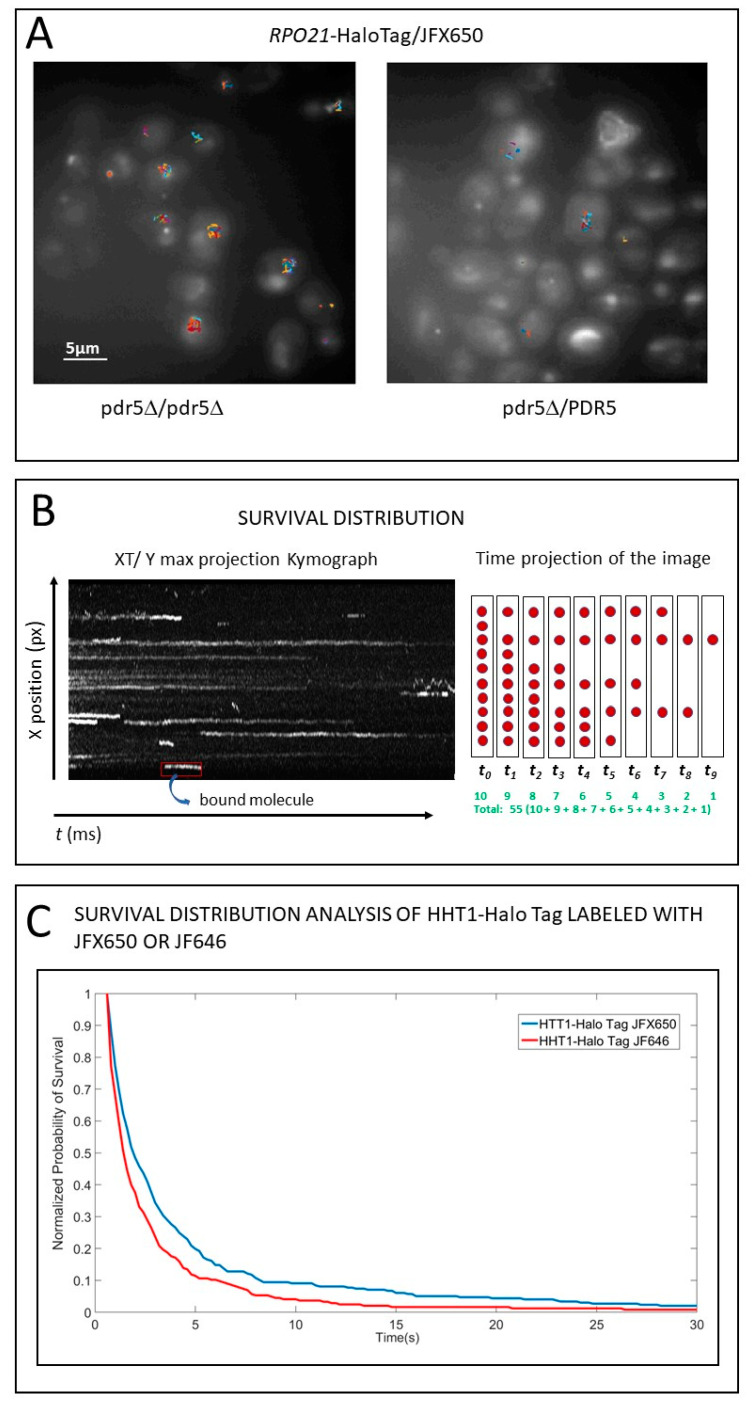
Staining for SMT in yeast. (**A**) Optimization of the dye retention by KO of the multidrug resistance transporter *PDR5*. A HILO-illuminated field of view for *pdr5* homozygotes (left) and heterozygotes (right) stained for *RPO21*-HaloTag (the largest subunit of RNA Pol II) with 10 pM of JFX_650_ ligand. Traces of single molecules are superimposed on the first frame of the movie. (**B**) Molecule survival in the field of view. Left: kymograph (X/T/Maximum intensity Y projection) of a single movie containing 10 nuclei of *HHT1*-HaloTag (yeast histone H3 protein) labeled with 1 pM of JFX_650_. Red rectangle with blue arrow indicates a bound molecule on the kymograph. Right: Schematics of survival of the molecules (red circles). Ten molecules are identified in frame *t*_0_. They may disappear in subsequent frames due to photobleaching or diffusion. The number of molecules visible at each time point may be plotted over time after normalization. (**C**) The choice of dye impacts the length of observation. Survival distribution for *HHT1*-HaloTag stained with 1 pM of JFX_650_ (blue) or JF_646_ (red) ligand. The curves are normalized to the total number of molecules observed in the first frame, and thus they are not dependent on the initial concentration of labeled molecules. The total number of HHT1p molecules detected by JFX_650_ Halo Tag ligand was 9214 and 5179 by JF_646_ per 149 nuclei.

**Figure 2 ijms-23-15895-f002:**
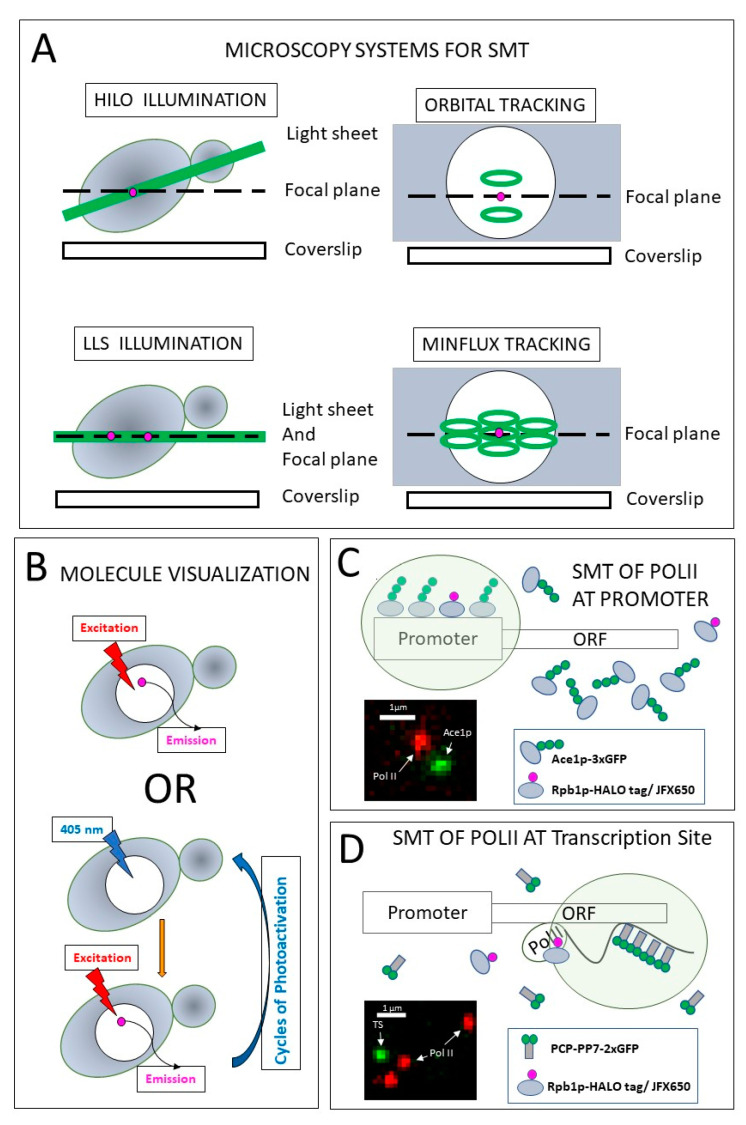
Imaging fluorescent molecules. (**A**) Schematics of microscopy systems for SMT. HILO and LLS illumination increase SNR. Orbital tracking follows molecules in 3D with two-color imaging; MINFLUX increases spatiotemporal resolution and allows 2D and 3D imaging in a single-color channel. (**B**) Schematics of molecule visualization ensure that few molecules at a time are detected. Top: In a sparsely labeled nucleus, single molecules may be excited directly. Bottom: In a nucleus densely labeled with photoactivatable fluorophores, a fraction of the molecules is activated by the appropriate laser line and then excited for fluorescence; after photobleaching of activated particles, the cycle may be repeated. (**C**) Schematics of tracking of RNA Pol II at specific promoter. Insert: frame of the movie, *RPO21*-HaloTag/JFX_650_ (red) and Ace1p-3xGFP bound to promoter of *CUP1* (green). (**D**) Schematics of tracking of RNA Pol II at specific TS. Insert: frame of the movie, *RPO21*-HaloTag/JFX_650_ (red) and PCP-PP7-GFP bound to stem-loops of the nascent mRNA of the single-copy reporter inserted into *CUP1* locus (green).

**Figure 3 ijms-23-15895-f003:**
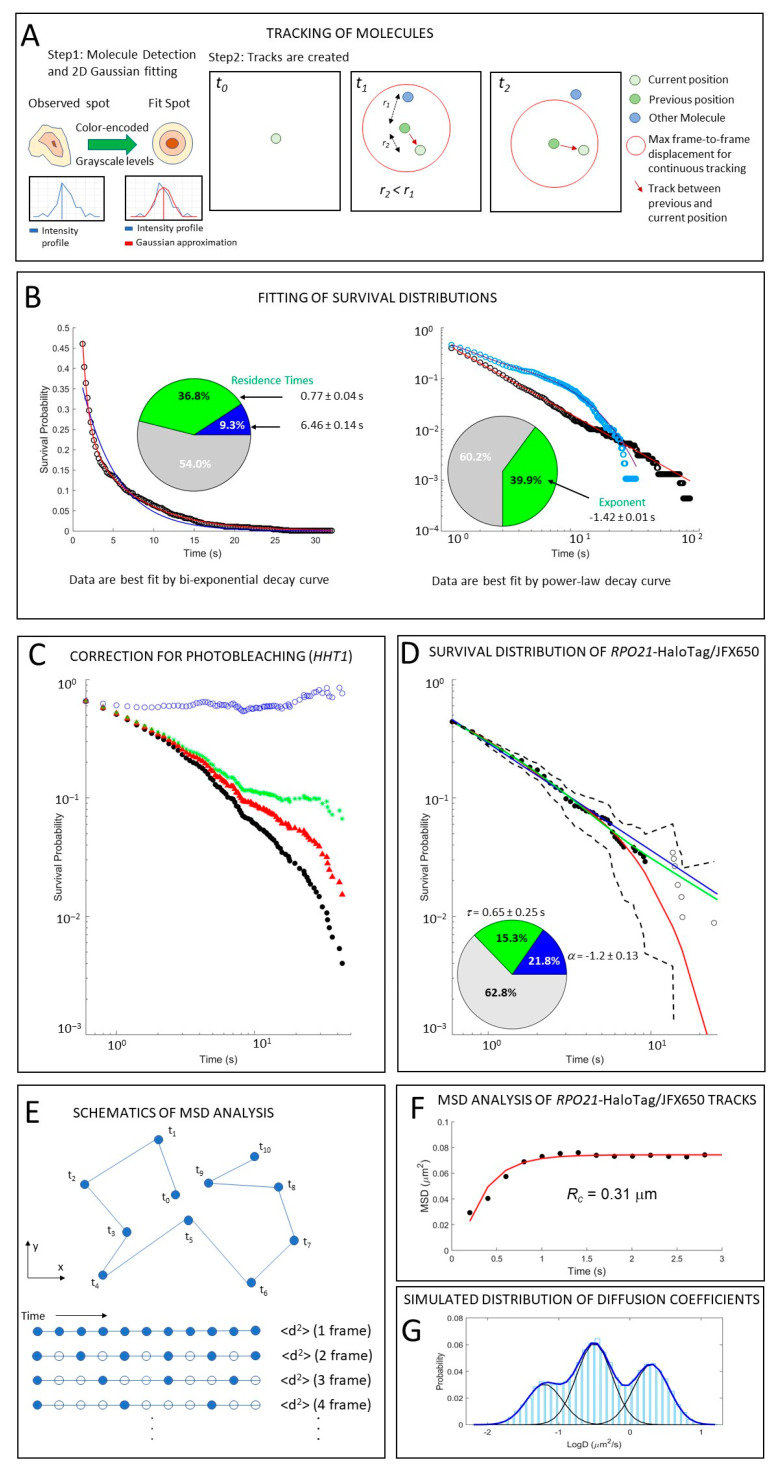
How to analyze the data? (**A**) Schematics of tracking single molecules. Step 1 (left): Detection of the fluorescent spot location by a Gaussian approximation of the real single molecule spot brightness distribution. Step 2 (right): Connecting two sequential positions of the same molecule in the movie (right). To link sequential positions of the same molecule observed in *t*_0_ and *t*_1,_ the shortest distance from the molecule’s position in *t*_0_ is selected. (**B**) Schematics of survival distribution analysis from simulated data. Left: Survival distribution (circles) approximated by a mono-exponential (blue line), or bi-exponential (red line) decay curve, and pie chart for estimated distributions of the diffusing sub-fraction (gray), short-binding events (green), and long-binding events (blue). Fitting with a bi-exponential decay also provides estimates of the residence times (τ) for the two binding sub-populations. Right: Log plots of bi-exponential (simulated data as blue-edged circles and fit as a gray line) versus Power Law (simulated data as gray-edged circles and fit as a red line) decay curves. Pie charts show estimates for the diffusing and bound fraction from the Power Law distribution (no τ may be estimated). Panel (**C**) Log-log plot of empirical survival distribution for histone H3 (*HHT1*-HaloTag/JFX_650_) with no photobleaching correction (black circles: experimental data), compared to the same survival distribution corrected for photo-bleaching correction with three different methods: intrinsic (red triangles); by decay of immobile H3 fraction (green asterisks), by full H3 decay (blue circles). The correction by the immobile H3 fraction (green asterisks) is deemed best, as it best represents what is observed: some histones are stably bound (corresponding to the plateau in the survival curve), while other histone molecules are unstably bound. Movies for *HHT1*-HaloTag/JFX_650_ molecules were collected with laser and camera exposures of 100 ms and 200 ms time intervals. A total of 807 tracks consisting of 17,504 molecular detections were analyzed from 278 cells. (**D**) Log-log plot of empirical survival distribution for RNA Pol II (*RPO21*-HaloTag/JFX_650_), with photobleaching corrected using the decay of immobile H3. The corrected curve (black circles) was fit with a bi-exponential distribution (red line), a pure Power Law (blue line), and a mixture of Power Law and exponential (green line). Based on the Bayesian Information Criterion for each fit, the Power Law/exponential function mixture (green line) was the best fit. Dashed black lines indicate 95% confidence intervals on the corrected empirical survival curve. Pie chart: diffusing fraction (gray), short-binding fraction (green) obtained from exponential portion of the fit, and slow-binding fraction (blue) obtained from Power Law fit. Movies for *RPO21*-HaloTag-JFX_650_ molecules were collected with laser and camera exposures of 100 ms and 200 ms time intervals. A total of 704 tracks consisting of 7748 detected particles were analyzed from 98 cells. (**E**) Schematics of MSD (Mean Squared Displacement) Analysis. Top: Track of a molecule in confined space, with equal time intervals between sequential positions. Bottom: Time intervals of the time-lapse are represented by circles. Average displacement is calculated for the different time intervals designated by solid circles. Panel (**F**) MSD and calculation of radius of confinement. Average displacement values are plotted against the time intervals used for their calculation. This distribution plateaus if the limits of confinement are reached. MSD for Pol II (*RPO21*-HaloTag/JFX_650_, collected with laser and camera exposure of 100 ms and 200 ms time intervals (black dots), and fit (red line) to obtain the radius of confinement (*R_c_*). The same 704 tracks used to calculate the survival curve in (D) were used to calculate the MSD. Panel (**G**) Simulated distribution of diffusion coefficients. Logarithm of diffusion coefficients plotted for a group of simulated tracks (blue bars). The distribution for the whole population of simulated tracks is then fit to a mixture of three Gaussian distributions (black curves), the sum of which is the blue curve. This simulation shows a three-state distribution with slow (left peak), intermediate (middle peak), and fast (right peak) motion.

**Figure 4 ijms-23-15895-f004:**
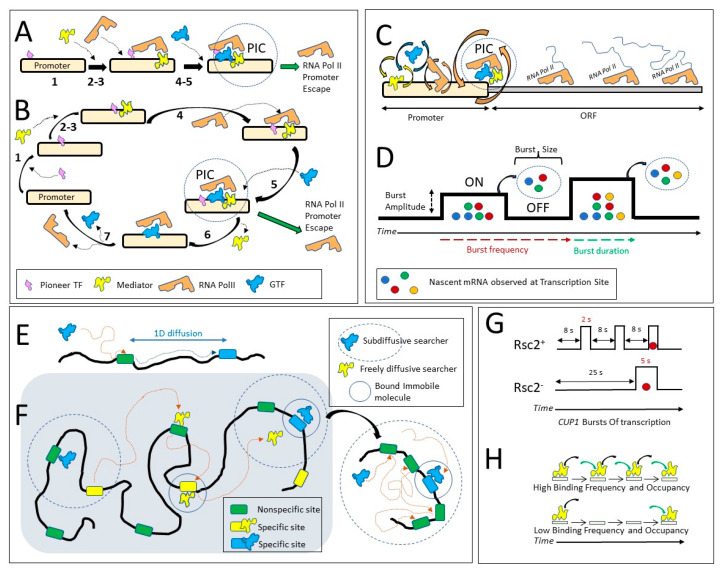
Lessons from SMT. (**A**) Sequential formation of PIC, based on ChIP. Steps 1–5 schematically depict sequential formation of PIC, which is then followed by RNA Pol II escape from PIC and elongation. (**B**) Schematics depict cycles of binding of pioneer factors, Mediator, GTFs, RNA Pol lI. Binding of the pioneer factor necessary for the promoter opening by chromatin remodelers is followed by multiple cycles of PIC assembly (1–5) and disassembly (6–7). Alternatively, assembly of PIC may be followed by RNA Pol II escape and elongation. (**C**) Schematics of the fast cycles of PIC formation followed periodically by RNA Pol II promoter escape and elongation; multiple polymerases may transcribe the same ORF at the same time. (**D**) Burst parameters for the gene bursting activity over time, where periods of inactivity (OFF) are followed by periods of activity (ON). In these schematics, individual molecules of mRNA presented by color-coded balls are released before the burst ends. Multiple nascent mRNAs may be present at the same ORF, as depicted by the simultaneous presence of two balls of different colors for the first burst or three for the second burst. (**E**) Schematics of 1D diffusion. Protein binds to the nonspecific site and proceeds to specific sites by sliding along the DNA. (**F**) Nuclear space exploration by proteins searching for specific sites (yellow for yellow molecule, blue for blue molecule) and sampling nonspecific sites (green). Non-compact non-constrained searchers move by isotropic diffusion, explore relatively large areas, and perform non-redundant search, leaving many targets unsampled (yellow molecule). Zones of restricted mobility to which proteins may be confined (blue molecule of the cartoon) are designated by dashed circle lines. Compact searchers (blue molecule) sample multiple binding sites performing redundant search as depicted in magnified constrained area pointed to by arrow. (**G**) RSC complex is necessary for high frequency of Ace1p binding and for the reduction of Ace1p residence time at *CUP1* promoter. Burst dynamics over time are represented by black line. Red circle represents nascent mRNA at the promoter. (**H**) High binding frequency (TF on, green arrow; TF off, black arrow) leads to high time-averaged occupancy of the promoter over time by TF (yellow molecule) (Top). Low binding frequency leads to low time-averaged occupancy (Bottom).

## Data Availability

Not applicable.

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
