# Peer review of "Impact of *Saccharomyces cerevisiae* on the Field of Single-Molecule Biophysics"

_ijms, 2022, doi:10.3390/ijms232415895_

Round 1
Reviewer 1 Report
Overall, this is a very comprehensive review of SMT in yeast. This is by far one of the most detailed reviews of transcription dynamics at the SM level that this reviewer has seen in the past several years.
A few areas the authors should address:
One minor issue is that the title implies a more general review of SM biophysics in yeast. This reviewer’s impression is that the review as written is more about single molecule approaches to transcription in yeast - perhaps the authors can expand some sections to provide a few examples of how the approaches used in transcription can be generalized to study other processes at SM level.
In Section 2.4, the authors address advantages/disadvantages to using yeast - authors should consider adding the ability to use mutants that will delay at specific cell cycle stages and availability of conditional alleles of numerous proteins to determine functional impact.
The authors do a very good job of highlighting some of the differences in transcriptional dynamics between mammalian and yeast - authors should consider adding that some discrepancies may in part be due to a lack of DNA methylation and the corresponding factors that recognize methylated DNA in yeast; another is yeast lack of some histone modifications and histone variants found in mammals.
Reviewer 2 Report
In this manuscript, authors summarize lots information of single molecule tracking that is a nice work. It includes what, how, why is single molecule tracking and its application in not only in mammalian cells but also in yeast cell. At last, authors also discuss the relative future prospects.
Overall, authors did a good job to make this manuscript, it organizes well and describes clearly enough, but in this review type manuscript, according to the title, it should focus on yeast, but go through all manuscript, not only limited in introduction, authors expend to other cell fields in almost every main section. I wonder whether authors can trim their manuscript to more focus on yeast as title point out or authors can modify the title to fit main text much more. And this is a review type manuscript, I do not see any figure/table regarding to authors’ “real experiment result”. I confuse what are the methodology, software, validation…. of author contributions and what do their means for? Another serious minor comment: a lot of references are not cited completed, the majority are missing or not correct page(/volume), such as reference 7, 8, 13, 14……. Authors have to carefully check and correct all the reference.
